# Touch or click friendly: Towards adaptive user interfaces for complex applications

Ibrar Hussain[1], Iftikhar Ahmed Khan[1]*, Waqas Jadoon[1], Rab Nawaz Jadoon[1], Abdul Nasir Khan[1], Muhammad Shafi[2]

1 Department of Computer Science, COMSATS University Islamabad, Abbottabad Campus TOBE Camp, Abbottabad, Pakistan, 2 Faculty of Computing and Information Technology, Sohar University, Sohar, Oman

* iftikharahmed@cuiatd.edu.pk

**Data Availability Statement:** All relevant data are within the manuscript and its Supporting information files.

**Funding:** The author(s) received no specific funding for this work.

## Abstract

This study evaluated the usability of a direct manipulation device (touchscreen) vs. indirect manipulation devices (mouse and touchpad) on the selected Microsoft (MS) Word tasks as per ISO-9241-11 standard. MS Word was taken as an example of a complex application. The tasks were evaluated in terms of touch-friendly or click-friendly using efficiency, effectiveness, and satisfaction parameters to propose a customized task menu. The experiment was conducted with fifty-four participants, divided into three MS Word usage-based expertise groups. Each participant performed fifty-six tasks using a mouse, a touchpad, and a touchscreen. To assess task-level usability, individual one-way ANOVAs were performed for each task to gauge both efficiency and effectiveness. It's worth noting that the touchscreen significantly outperformed other input methods in just one specific task regarding effectiveness. Consequently, an ANCOVA was employed, with task completion time as the independent variable and the number of errors as a covariate, to further investigate effectiveness. A total of 19 (34%) of the total tasks were found to be significantly efficient with a mouse, while 21 (37.5%) were significantly efficient with a touchscreen. Based on the results, a customized menu is recommended for MS Word-like applications that combine actions in touch-friendly tasks and mouse-friendly tasks separately.

## 1. Introduction

A meaningful interaction between a user and a computer always involves a user goal. The goal is achieved by a set of tasks which in turn are composed of actions. Rogers et al., [1] defined action as a singular user-initiated operation or gesture, such as clicking a button, dragging, and dropping a file, or typing a keystroke. Actions are typically smaller, more granular interactions than a task. A task often involves a sequence of actions and interactions. It represents a more significant user goal, such as composing an email, editing a document, or purchasing online. Users take actions or perform tasks using different input devices, of which the most used point-and-click oriented are computer mice, touchpads, and touchscreens.

Each input device, whether a touchscreen or a mouse, offers a variable user experience which may lead to different results for similar actions or tasks [2, 3]. Comparative studies between mice and touchscreen found mice more effective [4]. However, some studies such as

**Competing interests:** The authors have declared that no competing interests exist.

Wood et al. [5] showed that touchscreen input is fast and preferable for actions like icon selection on low-resolution screens. They concluded that a touchscreen is good for novice users while a mouse is better for expert users. In context, there is no conclusive evidence in favor of any single point-and-click device that generally could be considered superior in executing every action or task.

The efficiency and effectiveness of performing tasks could further deteriorate with an increase in the complexity of an application. The term complexity here means the required number of tasks that can be performed on an interface. MS Office applications like MS Word could be categorized as complex due to the availability of more than three hundred unique tasks [5]. Some studies evaluated the usability of MS Word input actions. For example, the usability of actions such as scrolling, clicking a button, or text selection was measured by Zabriskie et al.[6]. Some literature such as [7, 8] associated actions with the most suitable devices to execute those actions. For example, scrolling is touch-suitable while text selection is click-suitable. A recent doctoral thesis by Harding [9] compared different word editor applications' usability on different mobile devices. However, to the best of the authors' knowledge, no study has measured the task-level usability of complex applications like MS Word.

The evaluation of task-level usability over action-level usability becomes important with the increasing complexity of the applications. The action-level device suitability might not be useful at the task level because a task may include many actions. For example, a task having five actions could be composed of three touch-suitable actions and two mouse-suitable actions. Switching devices as per action suitability within a task is not practical, especially when those actions are not consecutive. Some studies such as [10, 11] designed 12 and 15 tasks for their prototypes to evaluate their usability. However, the study did not consider real-world applications. Secondly, the tasks were also limited in number. Researchers like Lindgaard & Chattratichart [12] suggested that the design of the tasks, the number of tasks, and their coverage also need to be researched for their role in the usability tests. Therefore, this research aims to measure the qualitative and quantitative usability of MS Word tasks using three input devices, i.e., mouse, touchpad, and touchscreen.

Besides, this study selected fifty-six most common tasks of MS Words with mixed difficulty levels, i.e., easy, medium, and difficult. This study also added rigor to the analysis by examining the task-level usability of MS Word using a sizeable number of participants as well as tasks. The results of this study could help to design a customized menu by grouping tasks based on the most suitable input device. Moreover, the customization of a menu could further help to reduce users' input time on the application interface.

All the related literature either lacks testing three input devices together or lacks in using many tasks of mixed difficulties, or many users of mixed skills. Therefore, the question arises whether another study with all such requirements is necessary. For example, can the result of comparing three devices A, B & C be implied from two studies comparing A & B and B & C? Moreover, is it possible that no previous experiment had mixed task difficulties because such a requirement is not important or can be otherwise implied from separate studies?

The question is answered by Kempf-Leonard [13] in the following words: "The relationship between one measure and another may be a true relationship, or it may be a spurious relationship that is caused by an invalid measurement of one of the measures. That is, the two measures may be related because of improper measurement, and not because the two measures are truly correlated with one another. Similarly, two truly related measures may remain undetected because an invalid measurement prevents the discovery of the correlation." As all measures are not perfectly valid, further experimentation should be conducted to find a direct relationship and therefore another motivation for this work.

The rest of the paper is organized as follows: Section 2 discusses related work and compares touchscreen, mouse, and touchpad for input. Section 3 discusses the preparation of materials for the study, while Section 4 discusses the experimental setup. Section 5 reports the results and discussions. The last section concludes the paper, highlights some of the limitations in the work, and points out some future directions to extend this work.

## 2. Related work

As highlighted in the introduction section, the research aims to measure task-level usability of complex applications like MS Word while using any of the three input devices (Mouse, touchscreen, and touchpad). Usability is defined as the ease with which a user can use an interface [14]. ISO standard 9241–11 [15] defined usability as "The extent to which a system, product, or service can be used by specified users to achieve specified goals with effectiveness, efficiency, and satisfaction in a specified context of the use". Efficiency is defined by the standard as "resources used for the results achieved". The resources here in this study include time used to complete a task. Effectiveness is defined by the standard as "accuracy and completeness with which users achieve specified goals". Accuracy is measured in terms of errors made while performing a task. User satisfaction is defined as "The extent to which the user's physical, cognitive, and emotional responses that result from the use of a system, product, or service in meeting the user's needs and expectations" and is measured with a System Usability Scale (SUS) [16]. Three performance measures will be utilized to compare devices and tasks. In addition, another aim is to add rigor compared to existing studies in terms of the number of participants used in the study, the number of tasks used in the study, mixed task difficulties (easy, medium, difficult) used in the study, and mixed types of users used in the study. For this purpose, 14 studies related to the tasks or actions usability were studied and their summary is presented in Tables 1 and 2 below.

Table 1 summarizes the related work by presenting the task types utilized and their performance measures. As can be seen from the second column of Table 1, "Task Type", the existing work has utilized multiple but simple tasks for the studies. Some studies such as Zabramski et al., [7], Zabramski et al., [6] and, Hooten et al., [10] used tasks such as drawing different figures. Some studies such as Rogers et al., [17] and Maclaughlin et al. [18] used selection controls such as selecting an item from a list box, drop-down list, or up/down button, etc. Wood et al., [5] used drag and drop of different objects. Steinert et al. [19], Baldus et al. [20], and Beelders et al. [21] utilized editing tasks in different contexts. However, it is clear from Table 2 that none of the studies used more than 15 tasks and all studies used custom-made applications in contrast to this study that utilized application in the real word use.

Furthermore, all the studies used either "error rate", "task completion time", "task completion rate" or "user satisfaction". The studies used different terms for user satisfaction such as user comfort or user experience. It is important to highlight the difference between task completion time and task completion rate. The former is about measuring the time taken by a task whereas the latter measures tasks completed out of total available tasks. In this context, 11 mentioned studies used error rate, 13 used task completion time, 5 used task completion rate and 6 used user satisfaction. This research utilized ISO Standard 9241–11 which explains all three performance measures i.e., efficiency (task completion time, task completion rate), effectiveness (error rate), and user satisfaction.

Each row of Table 1 shows the suggested preferred device of the corresponding study, the accurate device, and the device that is efficient as compared to other devices used in the study. In this context, some studies such as [5, 19, 20, 22, 24] suggested the mouse as the preferred device for doing the tasks. Some studies such as [6] and [17] suggested the touchscreen as the

**Table 1. An overview of the existing related studies in the context of task types and performance measures utilized.**

| Study | Task Type | Measure of Performance | Preference | Accurate | Efficient |
|---|---|---|---|---|---|
| Zabramski et al. [7] | drawing | Error rate, Time to complete the tasks | mouse/pen | Mouse/Pen | touchscreen |
| Rogers et al. [17] | use of list box, dropdown list box, up/down button, and text box | Task time, Error rate | touchscreen | Mouse | touchscreen |
| Wood et al. [5] | drag and drop | Error rate, Response time, user Experience | Mouse | Mouse | touchscreen |
| Zabramski et al. [6] | tracing, selecting, and steering tasks in MS Paint | Task completion time, accuracy | touchscreen | Mouse | touchscreen |
| Hooten et al. [10] | drawing | Input error, speed for drawing tasks | — | Mouse | touchscreen |
| Lane et al. [11] | cut, copy, paste, undo, redo, bold, italic, and underline | Task time | Keyboard | Keyboard | Keyboard |
| Maleckar et al. [22] | Web-based | Task Completion rate, User Experience | Mouse | Mouse | touchscreen vs. touchpad |
| Senanayake & Goonetilleke [23] | Steering | Error rate, User comfort | dependent on the task | Mouse | touchscreen |
| Steinert et al. [19] | login, writing a comment, returning to the main page, writing a message, adjusting settings, and starting an audio-video call | Task completion rate, Task Completion Time | Mouse | marginal differences | marginal differences |
| Travis et al., [24] | drag and drop, point, click, and contextualize | Task completion time, Error rate, and user Satisfaction | Mouse | Mouse | — |
| Maclaughlin et al. [18] | clocking on a list box, dropdown list box, up/down button, and text box | response time, accuracy | dependent on the task | — | — |
| Cockburn et al. [25] | tap, drag, and radial drag | target acquisition time and error rate | dependent on the task | faster for radial tasks | touchscreen for tapping |
| Baldus et al. [20] | Editing tasks in a moving environment | Task completion time, Error rate | Mouse | Comparable performance | Comparable performance |
| Beelders et al. [21] | word processing tasks using pictorial icons and then the same tasks using text buttons | Task completion time, errors made | — | No difference | No difference |

preferred device. However, studies such as [7] and [11] suggested multiple devices (i.e., mouse and pen or mouse and keyboard) as the preferred devices. Studies such as [18, 23, 25] did not suggest any preferred devices and mentioned that the choice of a device is dependent on the underlying task.

**Table 2. Shortcomings in the existing studies (✓ means a study fulfills a requirement and X means a study does not).**

| References. | Mouse, touchscreen, and touchpad comparison | Use of mixed tasks difficulty | No. of tasks | No. of users | Mix types of users | Usability measure | Recommendations |
|---|---|---|---|---|---|---|---|
| [2] | ✓ | × | 2 | 85 | ✓ | × | × |
| [6] | × | × | 1 | 16 | × | ✓ | × |
| [8] | × | × | 12 | 10 | ✓ | × | × |
| [11] | × | × | 15 | 6 | × | × | × |
| [17] | × | × | 6 | 24 | × | ✓ | × |
| [18] | ✓ | × | 1 | 18 | ✓ | ✓ | × |
| [22] | × | × | 6 | 32 | ✓ | ✓ | ✓ |
| [23] | × | × | 3 | 30 | × | ✓ | ✓ |
| [24] | × | × | 5 | 48 | × | ✓ | ✓ |
| [25] | × | × | 10 | 45 | × | ✓ | × |
| [26] | ✓ | × | 7 | 32 | × | ✓ | × |
| [27] | × | × | 1 | 18 | × | ✓ | × |
| [28] | × | × | 5 | 40 | × | ✓ | × |

The studies listed in Table 1 also show that the mouse was mostly accurate, and the touchscreen was mostly efficient. However, there are some variations as well. For example, the studies [19–21] either found no differences between different devices or found the performance comparable to each other. Similar results are found for efficiency as well. Table 1 also indicates which type of tasks can be executed with accuracy or efficiency with what type of device. For example, [5–7, 10, 17, 22–24] suggested that tasks such as drawing, use of a list box, selection from dropdown list box, selecting up/down button, steering tasks, drag and drop, point, click, tracing, and selecting tasks can be executed with more accuracy with a mouse and efficiently with a touchscreen. Some tasks' execution in terms of accuracy and efficiency are comparable when the selection of a device is a concern. For example, [19–21, 25] suggested that web-based tasks, editing tasks in a moving environment, word processing tasks by using pictorial icons or text buttons, tap, drag, and radial drag can be executed with any of the devices with significant differences.

Table 2 summarizes the significance of this study in the context of the related work. Table 2 indicates that only three studies, i.e., [2, 18, 26], compared three input devices (Mouse, Touchpad, and Touchscreen). Though the study [2] used more users (85) for evaluation as well as mixed expertise, however, the study utilized only 2 tasks compared to this study which provided additional rigor by using 56 tasks. The study [18] in addition to three devices, used a mix of user types. Furthermore, the study measured usability in terms of efficiency and effectiveness. Yet, the study used only 1 task, only 18 users, no mixed tasks difficulty, and provided no recommendations. The study [26] used only 7 tasks and 32 users. Furthermore, none of the studies used a mixed task difficulty and only one study [11] used the most, i.e., 15 tasks. The study [5] did not perform usability evaluations and did not provide recommendations as well.

## 3. Materials and methods

### 3.1. Selection of the participants

The study aimed to measure complex applications' point-and-click-based task usability to group them into touch-friendly tasks and click-friendly tasks. The purpose was to move toward adaptable user interfaces for complex applications that could minimize work completion time without errors. The point-and-click devices selected were the Mice, Touchpads, and Touchscreens. Furthermore, data collected from homogeneous types of users cannot be generalized [19], therefore, the participants were divided into three groups, i.e., experts, intermediate skilled, and novices based on MS Word usage. Similar categorization is used in the studies [5, 24, 29] as well. The participants with more than five years of experience were categorized as experts and up to five years of experience were categorized as intermediate skilled. The participants with no more than 6 months of experience with computers and MS Word were categorized as novice users.

The α error probability was taken as 0.05 and the power (1-β) error probability was taken as 0.95 for medium effect size. A priori repeated measures, within and between subjects' power analysis were conducted with the G*Power 3.0.10 tool. The sample size determined by the tool was 45. Nielsen and Landauer [30] suggest that 5–10 evaluators/test users could find 75% of the usability problems in medium-large software projects. Faulkner [30] suggested that fifty participants could help to find 100% of the problems. Therefore, an experimentation plan was devised to have sixty participants (20 in each of the expertise groups). The sample was a convenient sample formed of colleagues and students at the university and college of the authors. Hornbaek [31] showed that half of the studies of the ACM (Association of Computing Machinery) and CHI (Computer-Human Interaction) conferences used students as participants.

All participants were volunteers as there was no incentive to take part. To ensure an equal number of participants in each expertise group, novice users were selected from the first year of college. Intermediate and expert users were selected from the university, and colleagues of the authors. In the starting pilot phase of the study, six out of sixty participants (2 from each of the expertise groups) were not able to complete the tasks due to some problems in the application as their interaction-related recorded videos got corrupted. The problem was resolved. However, the data of the remaining fifty-four participants were only used. Out of fifty-four participants, twenty-nine were males and twenty-five were females with a mean age of 28 years and an age range from 18 to 42. All participants were right-handed with normal eyesight. As the tasks included selecting colors, therefore, participants were questioned about whether they experienced color blindness. Nevertheless, none of the participants described themselves as color blind.

## 3.2. Task selection

For the selection of the task, MS Word 365 was used. This version has approximately three hundred-plus tasks [32]. As a first step, the tasks that can be successfully performed only with a mouse, touchscreen, and touchpad were selected. The tasks like finding, commenting, and labeling that require a keyboard were not selected. Secondly, the tasks visible on the ribbon, after a new installation were only selected. The reason is that tasks or actions that are out of sight could be difficult to find and perform for novice users [21, 25, 32]. Among the tasks having a similar command structure (e.g., text alignment), only one of the tasks was selected (i.e., align Centre). The selected tasks were grouped as explained in the following section. Only fifty-six tasks met the criteria and were included in the study (c.f. S1 and S2 Appendices).

## 3.3. Grouping of the tasks

In the existing related literature, the chosen tasks were simple in structure. Most studies involved only actions like drag, drop, cut, copy, paste, and click. Very few studies used tasks composed of 3–4 actions. In this study, tasks with a minimum of three and a maximum of twenty-one actions were involved. Therefore, a criterion was needed to categorize the tasks into easy, medium, and difficult. The tasks were given a difficulty index based on the number of actions needed to complete a task. In some tasks, users were instructed to repeat the task and that repetition is considered one action. For example, task T32 is described as "By marking five words one by one (Campuses, Faculties, Academic department, Research Centre, students), insert the index after the table of figures. Hide the paragraph marker." Each selection of a word resulted in one action and marking each word resulted in another action and overall, in twenty-one actions. A similar method was also used by Beelders et al. [21] and Kortum et al. [33] though for different settings. Beelders et al. [21] used settings such as pre-task training and conducting the actual experiment after a few weeks. Whereas Kortum et al. [33] provided task T1 and then the SUS scale to rate usability, task T2 and SUS, and so on for the five tasks. Furthermore, the tasks were assessed as per the GOMS model introduced by Card et al., [34]. This was a specialized human information processor model that describes a user's cognitive structure on four components. A GOMS model is composed of Methods (M) that are used to achieve specific Goals (G). In this study, the goals to be accomplished and the final state presented by the user are provided in the scenario document (c.f. S4 Appendix). The Operators (O) are a set of operation sequences consisting of perceptual, motor, or cognitive behaviors as provided in S1 and S2 Appendices. The Method (M) refers to the rational order of goals and operations again evident from S1 and S2 Appendices, whereas Selection (S) refers to the reasonable operation method corresponding to the usage scenario that was provided through S4

Appendix. The GOMS model is still used in most of the latest research [35]. The instructions to perform the tasks were written down, printed, and provided to the users.

## 3.4. Scenario document

All tasks were performed on a given text in the form of a scenario. The text used was an existing text about the introduction of the authors' university and was adopted from the university website. The text was two pages in length (c.f. S4 Appendix). A similar procedure was also used by Tak et al. [36].

## 3.5. Questionnaire

A questionnaire was prepared in two parts. The first part was to collect demographic information like gender, age, language, qualification, handedness, etc. Information about the experiences of the participants with the mouse, touchscreen, touchpad, and MS Word was also collected in this part. The second part of the questionnaire (c.f. S3 Appendix) was used to collect user satisfaction with the use of input devices. The second part of the questionnaire was based on the SUS scale that was created by John Brooke in 1986. Vlachogianni and Tselios [37] indicated in their review paper that the SUS scale is a highly reliable and valid scale for studies with small sample sizes as well.

## 3.6. Equipment

A laptop with a touchscreen, touchpad, and mouse connectable to a USB port was used for the experiment. The laptops used in the experiment had the following specifications. CPU Intel Core M, 4GB RAM, 32GB ROM, 500GB HDD, and a ten-inches screen. The relevant software installed on the laptop was Microsoft Windows 8.1 Enterprise and MS Word 365. Camtasia software was installed and used to record on-screen activities.

## 3.7. Ethics

The Departmental Ethics Committee–called the Project Research & Evaluation Committee (PREC) reviewed and approved this study under approval number CUI-ATD/CS/PREC/003 dated 13 August 2020. A written consent form for the study was obtained from the participants. PREC approved the procedure and setup of the experiment. All participants consented to the experiment. All the data was collected between August 2020 to October 2020.

# 4. Experimental setup

In this experiment (schematic of experimental protocol—c.f. Fig 1), six laptops with touchscreens were used. As the keyboard was not part of the study, therefore, it was disabled and covered by paper to ensure it would not be used in any of the given tasks. Similarly, for touchscreen-only input, the touchpad was disabled. For performing tasks with a mouse, a USB mouse was attached, and the touchpad and the touchscreen were disabled. Furthermore, each participant was allotted a numeric ID to identify the screen recording along with their expertise level. The participants were not individually identifiable via this numeric ID or any other collected data.

The scenario text was copied to the desktop of all laptops in a notepad file. Participants were invited to the college of the first author for the experiment. All tasks were provided to the participants in printed form. The experiment was conducted in nine different sessions. Each experiment session accommodated six participants, three input devices, and fifty-six tasks. There was no introductory class for expert users. For the intermediate-skilled participants, an

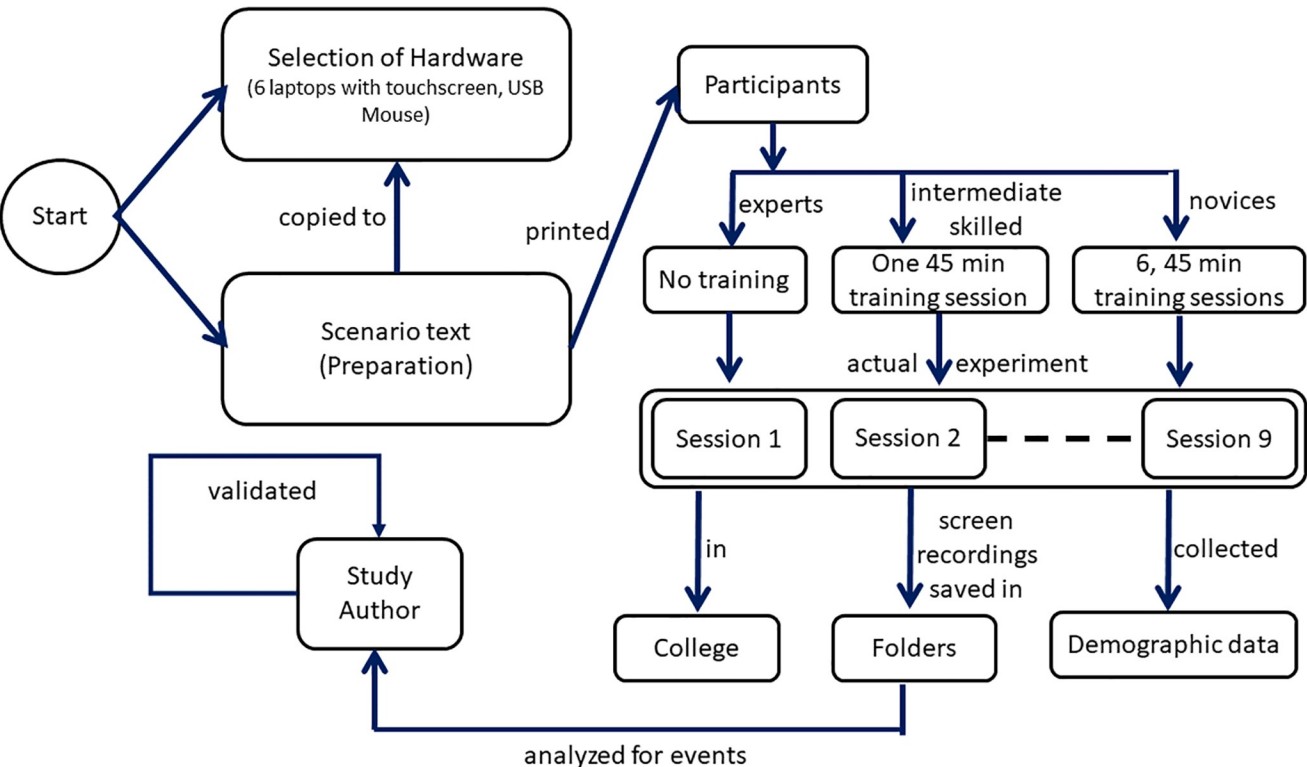

**Fig 1. A schematic of the experimental protocol.**

introductory class of 45 minutes was arranged to make them understand the tasks and the structure of MS Word. For novice users, six classes spread over six days, each of 45 minutes were conducted. In the two classes, novice participants were provided training with a mouse, in the next two days with a touchpad, and in the last two days with a touchscreen. A similar procedure was used by [29] as well. The users were given a list of tasks in a printed format and a solution to the tasks. Participants were told not to talk to each other during the experiment.

The study used a within-user design. Each participant completed each task three times alternatively with each of the input devices (mouse, touchpad, and touchscreen). To minimize the learning effect, the order of taking and completing the tasks was different for each user. As only six laptops of the required specification were available, therefore, the experiments were conducted in a group of six and with nine iterations to cover all participants. Each group of six had two experts, two intermediate skilled, and two novice users sitting alternatively to each other. Furthermore, the device type was also randomized.

To minimize boredom, participants could take a break of 5 to 15 minutes between the different input device phases of the experiment. The participants were free to do any activity during the break. The demographic data like gender, age, language, handedness, vision, color deficiency, and the number of hours per week they use a computer or MS Word was collected using a questionnaire after completion of the experiment.

The screen recordings were saved in separate folders. One of the study's authors analyzed the videos for the following events: task completed/ not completed, the time taken to complete a task, and the number of errors made during task completion. An error was considered if the participants selected a different action against the task specification. The information extracted was kept in an Excel sheet. Another author of the study validated the extracted information by

choosing and watching random 10 videos from the set of recorded videos of participants doing tasks in the experiment, and matching the data with the Excel sheet data.

## 5. Results

As already established, the study aims to measure the task-level usability of three point-and-click devices to suggest a customized menu for complex applications. The purpose of the following analyses is to cluster tasks for device-level usability and for a customized menu. In this context, the following three different analyses were conducted as per ISO 92411–11 [15] standard: 1) An analysis for efficiency 2) an analysis for effectiveness, and 3) knowing user satisfaction. However, the analyses could result in two different customizations: a customized menu for the efficient execution of tasks and a customized menu for the effective execution of the tasks. This in turn could be a cause of ambiguity for the end-users. Therefore, a third analysis is conducted taking efficiency as an independent variable and controlling the effect of errors (effectiveness) on the efficiency of the task.

### 5.1. Efficiency

To conduct the analysis, the mean task completion time was computed for all three groups of users with three input devices on each task. A One-Way ANOVA was conducted by taking task completion time as a dependent variable and the device type (Mouse, touchscreen, and touchpad) as an independent variable. The ANOVA test was conducted against each task separately. As Table 3 shows, out of fifty-six cases, 27 or 48% of the tasks' execution time was less with a mouse. Out of these 27 tasks, 21 (37%) tasks were executed with significantly less execution time. Furthermore, 3 tasks had a large effect size ($\eta2 \geq 0.14$), 8 tasks had a medium effect size ($\eta2 \geq 0.06$), and 10 tasks had a small Effect size ($\eta2 \geq 0.02$).

A total of 23 (41%) tasks had less execution time while using a touchscreen. Out of these, only fifteen tasks had significantly less execution time. Furthermore, 2 tasks had a large effect size ($\eta2 \geq 0.14$), 2 tasks had a medium effect size ($\eta2 \geq 0.06$), and 11 tasks had a small effect size ($\eta2 \geq 0.02$). The remaining 6 tasks had less execution time with a touchpad, but only two tasks had significantly less execution time with a medium effect size ($\eta2 \geq 0.02$).

### 5.2. Effectiveness

Another One-Way ANOVA was conducted by taking the mean number of errors made by the users on each task as a dependent variable and device type (Mouse, touchscreen, and touchpad) as an independent variable. As Table 4 shows, out of fifty-six cases, 32 or 57% of the tasks were executed with fewer errors with a mouse. Out of these tasks, a total of 25 (45%) tasks were executed with significantly fewer errors with a mouse. Furthermore, 5 tasks had a large effect size ($\eta2 \geq 0.14$), 7 tasks had a medium effect size ($\eta2 \geq 0.06$), and 13 had a small effect size ($\eta2 \geq 0.02$). A total of 22 (40%) tasks have fewer execution errors on a touchpad. Out of these, only sixteen tasks had significantly fewer execution errors with a touchpad. Furthermore, 4 tasks had a large effect size ($\eta2 \geq 0.14$), 8 tasks had a medium effect size ($\eta2 \geq 0.06$), and 4 tasks had a small effect size ($\eta2 \geq 0.02$). The remaining 2 tasks had fewer execution errors with a touchscreen, with only one task having significantly fewer execution errors with a touchscreen and with a medium effect size ($\eta2 \geq 0.02$). It is noticeable that people performed 97% of the tasks with fewer errors via pointing devices like touchscreen and touchpads. Whereas, for efficiency (Table 3) this ratio is approximately 60,40 in favor of the mouse. An interface is considered useable when it provides both efficiency and effectiveness on a single interface. To determine the usability considering both efficiency and effectiveness, a third analysis is conducted as illustrated in the following paragraph.

**Table 3. ANOVA for efficiency—The tasks, their F, p, uncorrected values, and their ES (effect size, η2).** In the ES column, *** is for a large ES, ** for a medium ES, and * for a small ES.

| Tasks efficient with a mouse | | | | Tasks efficient with a touchscreen | | | |
|---|---|---|---|---|---|---|---|
| Tasks | F | p uncorrected | η² | Tasks | F | p uncorrected | η² |
| T6 | 1.97 | 0.14 | *0.02 | T1 | 5.06 | 0.01 | *0.06 |
| T8 | 0.58 | 0.56 | 0.01 | T2 | 2.95 | 0.05 | *0.04 |
| T9 | 7.59 | 0.00 | **0.09 | T3 | 0.76 | 0.47 | 0.01 |
| T10 | 0.37 | 0.69 | 0.00 | T4 | 1.19 | 0.31 | 0.01 |
| T12 | 4.90 | 0.01 | *0.06 | T5 | 2.92 | 0.06 | *0.04 |
| T13 | 16.77 | 0.00 | ***0.17 | T11 | 3.25 | 0.04 | *0.04 |
| T17 | 1.29 | 0.28 | 0.02 | T14 | 3.90 | 0.02 | *0.05 |
| T18 | 3.93 | 0.02 | *0.05 | T15 | 1.12 | 0.33 | 0.01 |
| T19 | 3.48 | 0.03 | *0.04 | T16 | 3.59 | 0.03 | *0.04 |
| T20 | 1.79 | 0.17 | *0.02 | T22 | 1.14 | 0.32 | 0.01 |
| T21 | 7.06 | 0.00 | **0.08 | T24 | 2.55 | 0.08 | *0.03 |
| T23 | 14.21 | 0.00 | ***0.15 | T30 | 3.28 | 0.04 | *0.04 |
| T25 | 0.75 | 0.47 | 0.01 | T31 | 13.18 | 0.00 | ***0.14 |
| T28 | 1.94 | 0.15 | *0.02 | T32 | 0.88 | 0.42 | 0.01 |
| T33 | 1.50 | 0.23 | 0.02 | T38 | 7.53 | 0.00 | **0.09 |
| T34 | 4.32 | 0.01 | *0.05 | T40 | 1.69 | 0.19 | *0.02 |
| T35 | 9.82 | 0.00 | **0.11 | T43 | 0.52 | 0.60 | 0.01 |
| T37 | 7.13 | 0.00 | **0.08 | T47 | 0.58 | 0.56 | 0.01 |
| T39 | 7.98 | 0.00 | **0.09 | T48 | 1.70 | 0.19 | *0.02 |
| T41 | 12.95 | 0.00 | ***0.14 | T53 | 2.25 | 0.11 | *0.03 |
| T42 | 7.10 | 0.00 | **0.08 | T54 | 0.51 | 0.60 | 0.01 |
| T44 | 5.37 | 0.01 | **0.06 | T55 | 9.19 | 0.00 | **0.10 |
| T45 | 2.71 | 0.07 | *0.03 | T56 | 14.59 | 0.00 | ***0.16 |
| T46 | 3.01 | 0.05 | *0.04 | **Tasks efficient with a touchpad** | | | |
| T49 | 7.51 | 0.00 | **0.09 | T7 | 0.39 | 0.68 | 0.00 |
| T50 | 1.17 | 0.31 | 0.01 | T29 | 0.72 | 0.49 | 0.01 |
| T51 | 3.16 | 0.05 | *0.04 | T26 | 0.31 | 0.73 | 0.00 |
| | | | | T27 | 7.18 | 0.00 | **0.08 |
| | | | | T36 | 0.83 | 0.44 | 0.01 |
| | | | | T52 | 8.35 | 0.00 | **0.10 |

## 5.3. Efficiency with effectiveness as a covariate

A One-Way ANCOVA was conducted to suggest a customization of menus that considers both efficiency and effectiveness. Therefore, in this analysis, the independent variable taken was task completion time and the covariate variable was the number of errors in task execution. The dependent variables were device type (Mouse, touchscreen, and touchpad). As Table 5 shows, out of fifty-six cases, 27 or 48% of the tasks' execution time was lesser with a mouse. Out of these tasks, a total of 19 (34%) tasks were executed with significantly less time with a mouse. Furthermore, 4 tasks had a large effect size ($\eta2 \geq 0.14$), 8 tasks had a medium effect size ($\eta2 \geq 0.06$), and 7 tasks had a small effect size ($\eta2 \geq 0.02$). A total of 23 (41%) tasks were found to have less execution time on a touchscreen. Out of these, a total of twenty-one tasks had significantly less execution time on the touchscreen. Furthermore, 2 tasks had a large effect size ($\eta2 \geq 0.14$), 10 tasks had a medium effect size ($\eta2 \geq 0.06$), and 9 tasks had a small effect size ($\eta2 \geq 0.02$). The remaining 6 tasks had less execution time on a touchpad, with four tasks having significantly less execution time with a medium effect size ($\eta2 \geq 0.02$).

**Table 4. ANOVA for effectiveness—The tasks, their F values, p uncorrected values, and their ES (effect size).** in the ES column, *** is for a large ES, ** for a medium ES, and * for a small ES.

| Tasks effective with a mouse | | | | Tasks effective with a touchscreen | | | |
|---|---|---|---|---|---|---|---|
| Tasks | F | p uncorrected | $\eta^2$ | Tasks | F | p uncorrected | $\eta^2$ |
| T2 | 4.47 | 0.01 | *0.05 | T14 | 0.84 | 0.43 | 0.01 |
| T3 | 1.65 | 0.19 | *0.02 | T15 | 5.65 | 0.00 | **0.07 |
| T4 | 1.60 | 0.21 | 0.02 | Tasks effective with a touchpad | | | |
| T5 | 0.37 | 0.69 | 0.00 | T1 | 0.64 | 0.53 | 0.01 |
| T7 | 2.14 | 0.12 | *0.03 | T6 | 10.35 | 0.00 | **0.12 |
| T8 | 6.83 | 0.00 | **0.08 | T10 | 5.26 | 0.01 | **0.06 |
| T9 | 6.51 | 0.00 | **0.08 | T17 | 1.52 | 0.22 | 0.02 |
| T11 | 3.58 | 0.03 | *0.04 | T18 | 6.85 | 0.00 | **0.08 |
| T12 | 2.69 | 0.07 | *0.03 | T21 | 20.05 | 0.00 | ***0.20 |
| T13 | 61.07 | 0.00 | ***0.43 | T22 | 1.36 | 0.26 | 0.02 |
| T16 | 0.90 | 0.41 | 0.01 | T23 | 1.53 | 0.22 | 0.02 |
| T19 | 6.72 | 0.00 | **0.08 | T24 | 0.30 | 0.74 | 0.00 |
| T20 | 3.82 | 0.02 | *0.05 | T26 | 7.72 | 0.00 | **0.09 |
| T25 | 1.63 | 0.20 | *0.02 | T27 | 35.00 | 0.00 | ***0.31 |
| T28 | 5.13 | 0.01 | **0.06 | T36 | 1.48 | 0.23 | 0.02 |
| T29 | 5.69 | 0.00 | **0.07 | T38 | 2.52 | 0.08 | *0.03 |
| T30 | 4.95 | 0.01 | *0.06 | T41 | 6.02 | 0.00 | **0.07 |
| T31 | 6.54 | 0.00 | **0.08 | T42 | 12.98 | 0.00 | ***0.14 |
| T32 | 0.15 | 0.86 | 0.00 | T43 | 2.47 | 0.09 | *0.03 |
| T33 | 1.71 | 0.18 | *0.02 | T44 | 3.78 | 0.02 | *0.05 |
| T34 | 3.86 | 0.02 | *0.05 | T45 | 6.18 | 0.00 | **0.07 |
| T35 | 15.19 | 0.00 | ***0.16 | T47 | 17.71 | 0.00 | ***0.18 |
| T37 | 3.88 | 0.02 | *0.05 | T48 | 9.45 | 0.00 | **0.11 |
| T39 | 13.72 | 0.00 | ***0.15 | T50 | 6.85 | 0.00 | **0.08 |
| T40 | 1.32 | 0.27 | 0.02 | T54 | 1.68 | 0.19 | *0.02 |
| T46 | 20.28 | 0.00 | ***0.20 | | | | |
| T49 | 1.64 | 0.20 | *0.02 | | | | |
| T51 | 9.94 | 0.00 | **0.11 | | | | |
| T52 | 45.31 | 0.00 | ***0.36 | | | | |
| T53 | 0.34 | 0.71 | 0.00 | | | | |
| T55 | 1.01 | 0.37 | 0.01 | | | | |
| T56 | 3.01 | 0.05 | *0.04 | | | | |

Table 6 shows changes between an ANOVA conducted to measure efficiency and an ANCOVA conducted to measure efficiency with effectiveness data as a covariate. For example, although total tasks with less execution time stay the same for the mouse, the tasks with significantly less execution time are reduced from 21 (78%) to 19 (70%). For the touchscreen, the tasks with significantly less execution time increased from 15 (65%) to 21 (91%) tasks. Furthermore, with the touchpad, the tasks with significantly less execution time increased from 2 (33%) to 4 (67%).

Table 5, i.e., ANCOVA analysis formed the base for a possible customized menu suggestion. Therefore, the results in Table 5 are utilized to interpret Table 8 which shows the grouping of the tasks as per their difficulty index. The table also shows the possible grouping of tasks in the touch, mouse, and touchpad menus.

**Table 5. ANCOVA—The tasks, their F, p, uncorrected values, and their ES (effect size).** In the ES column, *** is for a large ES, ** for a medium ES, and * for a small ES.

| Tasks efficient with a mouse—errors controlled | | | | Tasks efficient with a touchscreen—errors controlled | | | |
|---|---|---|---|---|---|---|---|
| Tasks | F | p uncorrected | η2 | Tasks | F | p uncorrected | η2 |
| T6 | 5.320 | 0.01 | **0.06 | T1 | 6.08 | 0.00 | **0.07 |
| T8 | 1.97 | 0.14 | *0.02 | T2 | 6.18 | 0.00 | **0.07 |
| T9 | 30.41 | 0.00 | ***0.28 | T3 | 0.50 | 0.61 | 0.01 |
| T10 | 3.69 | 0.03 | *0.04 | T4 | 3.80 | 0.02 | *0.05 |
| T12 | 10.51 | 0.00 | **0.12 | T5 | 4.98 | 0.01 | *0.06 |
| T13 | 4.10 | 0.02 | *0.05 | T11 | 5.54 | 0.00 | **0.07 |
| T17 | 2.52 | 0.08 | *0.03 | T14 | 3.50 | 0.03 | *0.04 |
| T18 | 0.50 | 0.61 | 0.01 | T15 | 1.82 | 0.17 | *0.02 |
| T19 | 0.11 | 0.89 | 0.00 | T16 | 3.16 | 0.05 | *0.04 |
| T20 | 0.16 | 0.86 | 0.00 | T22 | 3.12 | 0.05 | *0.04 |
| T21 | 1.15 | 0.32 | 0.01 | T24 | 3.89 | 0.02 | *0.05 |
| T23 | 29.89 | 0.00 | ***0.27 | T30 | 8.02 | 0.00 | **0.09 |
| T25 | 0.14 | 0.87 | 0.00 | T31 | 10.40 | 0.00 | **0.12 |
| T28 | 0.54 | 0.59 | 0.01 | T32 | 1.59 | 0.21 | 0.02 |
| T33 | 0.71 | 0.49 | 0.01 | T38 | 16.17 | 0.00 | ***0.17 |
| T34 | 3.01 | 0.05 | *0.04 | T40 | 5.15 | 0.01 | **0.06 |
| T35 | 11.61 | 0.00 | **0.13 | T43 | 2.32 | 0.10 | *0.03 |
| T37 | 3.53 | 0.03 | *0.04 | T47 | 5.60 | 0.00 | **0.07 |
| T39 | 8.31 | 0.00 | **0.1 | T48 | 5.36 | 0.01 | **0.06 |
| T41 | 18.77 | 0.00 | ***0.19 | T53 | 5.73 | 0.00 | **0.07 |
| T42 | 7.87 | 0.00 | **0.09 | T54 | 1.70 | 0.19 | *0.02 |
| T44 | 13.28 | 0.00 | ***0.14 | T55 | 11.59 | 0.00 | **0.13 |
| T45 | 8.54 | 0.00 | **0.1 | T56 | 14.44 | 0.00 | ***0.15 |
| T46 | 12.55 | 0.00 | **0.14 | Tasks efficient with a touchpad with error-controlled | | | |
| T49 | 10.05 | 0.00 | **0.11 | T7 | 0.90 | 0.41 | 0.01 |
| T50 | 2.06 | 0.13 | *0.03 | T26 | 3.54 | 0.03 | *0.04 |
| T51 | 0.20 | 0.82 | 0.00 | T27 | 3.99 | 0.02 | *0.05 |
| | | | | T29 | 3.63 | 0.03 | *0.04 |
| | | | | T36 | 0.12 | 0.88 | 0.00 |
| | | | | T52 | 3.03 | 0.05 | *0.04 |

Moreover, 14 tasks in Table 3, 14 tasks in Table 4, and 12 tasks in Table 5 were not significantly different while performed with different devices. Even significantly different tasks had varying effect sizes as Table 6 indicates. Winer [35] suggested that: "The frequent use of the .05 and .01 levels of significance is a matter of convention having little scientific or logical basis.

**Table 6. Summary of the analyses.** TT = Total tasks, TC = Total change in effect size of tasks in percent, L = large, M = Medium, and S = Small effect size percentage.

| | | Tasks completed with a mouse | | | | | Tasks completed with a touchscreen | | | | | Tasks completed with a touchpad | | | | |
|---|---|---|---|---|---|---|---|---|---|---|---|---|---|---|---|---|
| Measure | Analysis | TT | L | M | S | TC | TT | L | M | S | TC | TT | L | M | S | TC |
| Efficiency | ANOVA | 27 | 11% | 30% | 37% | 78% | 23 | 9% | 9% | 48% | 65% | 6 | 0% | 33% | 0% | 33% |
| Effectiveness | ANOVA | 32 | 16% | 22% | 41% | 78% | 2 | 0% | 50% | 0% | 50% | 22 | 18% | 36% | 18% | 73% |
| Covariate | ANCOVA | 27 | 15% | 30% | 26% | 70% | 23 | 9% | 43% | 39% | 91% | 6 | 0% | 67% | 0% | 67% |

**Table 7. Statistics of one-way ANOVA.**

| Question No. | F (2,159) | p | μ of mouse | μ of touchscreen | μ of touchpad |
|---|---|---|---|---|---|
| 1 | 90.28 | < 0.001 | 4.8 | 2.7 | 2.9 |
| 2 | 49.28 | < 0.001 | 4.4 | 2.8 | 3 |
| 3 | 48.59 | < 0.001 | 1.37 | 3.4 | 2.8 |
| 4 | 64.73 | < 0.001 | 4.63 | 2.6 | 3.1 |
| 5 | 61.78 | < 0.001 | 4.59 | 2.6 | 2.8 |

When the power of the test is likely to be low under these levels of significance, and when Type I and Type II errors are of approximately equal importance, the .30 and .20 levels of significance may be more appropriate than the .05 and .01 levels". In this context, as per Table 5, 9 tasks out of 56 are above the benchmark of 0.30 and may not be suitable for the customized menu. However, these 9 tasks could be adjusted in a touch-friendly menu if most of the tasks are touch-friendly and vice versa.

## 5.4. User satisfaction

Participants completed a post-experiment questionnaire that collected opinions on the use of three input devices to complete the tasks. Participants were asked to rate the ease of use, their likes, dislikes, satisfaction, and dissatisfaction with each device using a 5-point Likert scale with 5 as "strongly agree" and 1 as "strongly disagree". A total of 5 questions (c.f. S3 Appendix) were asked on each device. A One-Way ANOVA was conducted taking each question as the dependent variable and the device type as a factor. The results showed significant differences in the subjective perception of the ease of use of each device for every question asked.

The result as per Table 7 shows that participants liked mouse interaction the most against every question asked. Furthermore, participants perceived the mouse as easier to use as compared to a touchscreen and touchpad. Comparable results were also found by Travis & Murano [24]. Despite the users' preference of a device (might be due to the comfort level or experience with a mouse), the empirical results reveal that task usability can be improved by introducing a customized touch-and-click friendly menu as illustrated in Table 8 below.

**Table 8. Grouping of the tasks according to the difficulty index in the context of Table 5.**

| Difficulty Index | Touch friendly tasks | | | | | Mouse friendly tasks | | | | | | Touchpad friendly tasks | | | Total |
|---|---|---|---|---|---|---|---|---|---|---|---|---|---|---|---|
| 4 | T3 | T11 | T54 | | | T8 | T10 | T13 | T25 | T39 | T46 | T7 | T36 | T52 | 12 |
| 5 | T2 | T30 | T43 | T53 | T55 | T20 | T42 | T44 | | | | | | | 8 |
| 3 | T1 | T56 | | | | T17 | T18 | T49 | T51 | | | T27 | | | 7 |
| 6 | T22 | T24 | T47 | | | T37 | T45 | T50 | | | | | | | 6 |
| 7 | T15 | T31 | T40 | | | T9 | T12 | T41 | | | | T29 | | | 6 |
| 9 | T4 | T5 | T14 | T48 | | | | | | | | | | | 5 |
| 11 | T16 | | | | | T19 | T35 | | | | | | | | 3 |
| 8 | | | | | | T34 | | | | | | T26 | | | 2 |
| 15 | | | | | | T21 | T23 | | | | | | | | 2 |
| 12 | T38 | | | | | | | | | | | | | | 1 |
| 13 | | | | | | T33 | | | | | | | | | 1 |
| 14 | | | | | | T6 | | | | | | | | | 1 |
| 20 | | | | | | T28 | | | | | | | | | 1 |
| 21 | T32 | | | | | | | | | | | | | | 1 |

## 5.5. Validity and reliability analysis

It is important to validate a finding against its claims. Furthermore, it is also important to generalize the results to other populations or settings. Therefore, for the validity analysis of this study, all necessary details related to the selection of the materials, criteria, and constructs are explained in the sections "Preparation of Materials" and "Experimental Setup". Furthermore, a covariate analysis (c.f. subsection 5.3) is conducted to evaluate the potential confounding factors affecting the tasks.

Reliability refers to the extent to which the study can produce consistent and reproducible results over time and across different samples or settings. The importance of conducting a reliability analysis is highlighted in [38]. To conduct a reliability analysis for the study, both inter-rater reliability and internal consistency reliability analyses were conducted. For the former analysis, intraclass correlations (ICC) were computed, while for the latter analysis, Cronbach's alpha was used.

To compute ICC, the effectiveness data of the tasks gathered from the participants of each expertise on each of the devices were combined in a single file. A similar process was conducted for efficiency data as well. The average measures were utilized for the analysis because of repeated measures of the same construct. Average measures can also help to reduce the measurement error and increase the precision of your estimates. The effectiveness resulted in an intraclass correlation of 0.860 showing excellent reliability. Similarly, the results of efficiency showed an intraclass correlation of 0.842 showing excellent reliability. Exactly similar results were obtained for Cronbach alpha showing high internal consistency.

## 6. Conclusion, limitations, and future work

In the context of usability matrices: effectiveness, efficiency, and user satisfaction, the results of this study are at par with related research. For example, this study found that the touchscreen is an input device that is not accurate and is the reason for making more errors with all the expertise levels. The results are in line with the research conducted by Maleckar et al. [22]. However, the touchscreen is fast, and novice users can perform better with the touchscreen as also stated by Zabramski et al. [6]. Intermediate skilled participants performed better with both a mouse and touchscreen; however, they were more accurate and efficient with a mouse. Expert users performed better with all three devices. A mouse was the most preferred input device in all three groups and a touchpad was the least preferred based on the subjective assessment of the participants, finding it in line with [8, 26, 30]. The reason for the preference might be because the users have more experience with the mouse, and they feel easier with a mouse.

Another finding of the study is that the use of the devices also depends on the tasks we are performing with a specific device. The study found that each device has different interactions for different tasks. Some tasks were performed better with the least preferred input devices. Therefore, we can conclude that a touchscreen can be used for some tasks more efficiently despite users' least preference. Besides, small targets and targets having less spacing in between are not touch-friendly tasks.

The overall conclusion of the study is that only a single device is not better for all tasks. Similarly, a single type of user is also not good with all types of devices and tasks. The optimality of the device depends on the nature of the tasks. These findings are also backed up by the literature which shows that some specific tasks such as drag tasks are better with a touchscreen, whereas a mouse is a suitable choice for tasks like scrolling [25]. As can be seen from Table 6, tasks executed better with a mouse can be combined with the tasks executed better with a touchpad. Furthermore, effectiveness has very limited tasks inclined towards a touchscreen and is highly inclined towards click friendliness. Both a mouse and a touchpad are click-

friendly, therefore, variation of customization can be better achieved by considering the efficiency with effectiveness as a covariate. Both the touchpad and mouse are similar in the context that both are indirect manipulation devices requiring a pointer to execute actions. Therefore, the touchpad and mouse could be combined into a single menu. A probable grouping of the tasks into menus is provided in Table 8 below.

As it is clear from Table 8, 23 tasks are touch-friendly, and 33 tasks are mouse/touchpad-friendly. Both menus contain tasks of difficulty index up to 9 and almost in equal numbers, i.e., 20 vs. 24 tasks, respectively. However, the tasks with a difficulty index above 9 mostly are grouped in mouse/touchpad-friendly menus with a touch-vs mouse ratio of 1:3. In this context, another significant conclusion of this study is that the tasks with a higher difficulty index are better executed with a mouse. The customization of MS Word 365 can be done by visiting File-Options and selecting the Customize ribbon tab. One can follow the URL [39] as well to do the customization.

The grouping of the tasks has its limitations. For example, although the devices might work well on the tasks listed in this research, the interface needs to be modified considering the current findings combined with maintaining the existing interface to maintain consistency. This would put extra constraints on the designers to group tasks based on touch or click friendliness. However, a deeper look at the tasks and their composition of actions might enable designers to design such tasks and thus can be considered as future work.

Another limitation of the study is not having enough participants (54) who were used for research. Some research literature on usability shows that the number of users above 50 is sufficient to find 100% usability problems. However, the division of the users into three groups could have an impact on the effect size of the study. Furthermore, this categorization was done based on their experience with input devices and MS Word, and their level of expertise was not measured objectively. However, the step was taken to generalize the usability of devices for all types of users. Another limitation of this study is related to MS Word Tasks, which have more than three hundred tasks. However, for this study, only fifty-six tasks were included. Therefore, in future studies, all tasks including keyboard input tasks could also be used. In addition, all tasks with a particular device usability could be grouped and could be evaluated to determine the interfaces' usability.

## Supporting information

**S1 Appendix. The tasks list.**
(DOCX)

**S2 Appendix. The task list with details for execution of each task.**
(DOCX)

**S3 Appendix. Questionnaire and modified SUS Questionnaire.**
(DOCX)

**S4 Appendix. Scenario document.**
(DOCX)

**S5 Appendix. Expertise based analysis.**
(DOCX)

**S1 Data.**
(SAV)

**S2 Data.**

(SAV)

**S1 File.**

(SAV)

## Author Contributions

**Conceptualization:** Iftikhar Ahmed Khan.

**Data curation:** Ibrar Hussain.

**Formal analysis:** Iftikhar Ahmed Khan.

**Funding acquisition:** Muhammad Shafi.

**Investigation:** Ibrar Hussain, Iftikhar Ahmed Khan, Waqas Jadoon.

**Methodology:** Ibrar Hussain, Waqas Jadoon, Abdul Nasir Khan.

**Project administration:** Waqas Jadoon.

**Resources:** Rab Nawaz Jadoon, Abdul Nasir Khan.

**Software:** Ibrar Hussain.

**Supervision:** Iftikhar Ahmed Khan, Muhammad Shafi.

**Validation:** Ibrar Hussain, Rab Nawaz Jadoon, Abdul Nasir Khan.

**Visualization:** Waqas Jadoon.

**Writing – original draft:** Ibrar Hussain.

**Writing – review & editing:** Iftikhar Ahmed Khan, Rab Nawaz Jadoon, Abdul Nasir Khan, Muhammad Shafi.

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
