## [Decision Letter · Decision Letter 0]

2 Oct 2023

PONE-D-23-26336Touch or Click Friendly: Towards Adaptive User Interfaces for Complex ApplicationsPLOS ONE

Dear Dr. Khan,

Thank you for submitting your manuscript to PLOS ONE. After careful consideration, we feel that it has merit but does not fully meet PLOS ONE’s publication criteria as it currently stands. Therefore, we invite you to submit a revised version of the manuscript that addresses the points raised below by the reviewers.

We look forward to receiving your revised manuscript.

Kind regards,

Rajagopalan Srinivasan

Academic Editor

PLOS ONE

Journal Requirements:

Reviewers' comments:

Reviewer's Responses to Questions

**Comments to the Author**

1. Is the manuscript technically sound, and do the data support the conclusions?

Reviewer #1: Partly

Reviewer #2: Partly

2. Has the statistical analysis been performed appropriately and rigorously? 

Reviewer #1: I Don't Know

Reviewer #2: Yes

3. Have the authors made all data underlying the findings in their manuscript fully available?

Reviewer #1: Yes

Reviewer #2: No

4. Is the manuscript presented in an intelligible fashion and written in standard English?

Reviewer #1: Yes

Reviewer #2: No

5. Review Comments to the Author

Reviewer #1: I have attached a file with comments organized by section for ease of review. The objective and applicability of this study are not clearly communicated. The relevance of the statistical analysis is therefore difficult to confirm. This study appears to aim to synthesize a number of prior studies, but some of the key variables and metrics are not clearly communicated.

Reviewer #2: The paper evaluates the usability of different input devices such as mouse, touchpad and touchscreen in performing a wide range of tasks in MS Word. The task-level usability is evaluated in terms of parameters that include efficiency, effectiveness and user satisfaction in line with ISO-9241-11 standard recommendations. Accordingly, the tasks have been categorized into touch-friendly and click-friendly. Further, the paper considers participant groups with multiple levels of expertise to ensure generalizability of results. The tasks considered are also of mixed difficulty levels. The work has the potential to pave the way for improving user interface design of complex applications by grouping tasks based on the most suitable input device. However, the paper can only be considered for publication after the following comments are thoroughly addressed.

1. The authors should add more recent literature in Section 1 (Introduction) and 2 (Related Work).

2. The way the existing literature has been reported in Related Work needs to be improved. It would be helpful if conclusions/insights derived from those studies are reported (briefly) so that the reader benefits from reading it. Just reporting the number/gender of participants, number of tasks, and what has not been done in the previous studies is not much helpful.

3. The authors may want to include inferences drawn from studies [9], [17], [24] as these studies have considered mouse, touchscreen and touchpad comparison [as depicted from Table 1]. Further, how these conclusions relate/align/ with the findings of this paper.

4. Line 238: …, all tasks were assessed as per the GOMS method. Assessed for what? L239-L244 may be suitably modified to make it evident and highlight its usefulness (conceptually).

5. Was there any specific reason that MS Word 2010 was used?

6. A schematic of the whole experimental protocol would be more beneficial than Figure 1 (which currently includes a picture of a laptop).

7. L339-L341: “A compromise power analysis for (N=60, df =2, No. of Groups=3, effect size= Medium) revealed a power of 0.73.” What does this signify/imply?

8. There is a mismatch between the title of Section 4.3 (L384) and L387. While the title mentions efficiency as covariate, the text in L387 considers effectiveness as a covariate. Also, the title of Section 4.3 is a bit confusing and may be modified suitably. Further, what is the rationale for the choice of a particular covariate measure?

9. Table 5 reports numbers. It may be more useful if the numbers are reported as percentages so as to make interpretation more efficient. Also, in the main text of the “Results” section, the percentages should also be provided consistently for ease of interpretation.

10. If possible, the authors may also want to report the summarized results of efficiency and effectiveness of various input devices for different expertise groups. This will help understand whether there is any impact of expertise on usability of devices, at least qualitatively.

11. Section 4.5. Please improve for better clarity especially the sentences related to validity. Also provide a conceptual significance of Inter-rater reliability and internal consistency, or at least refer the reader to a suitable reference.

12. L434-435: “Furthermore, a covariate analysis is conducted as explained in subsection J of Section IV to evaluate the potential confounding factors affecting the tasks” Subsection J does not exist.

13. L476: “with a touch-to--to-mouse/touchpad number 3 vs. 9 or with a ratio of 1:3” is not clear.

14. L 478: “The customization can be done as per the suggestion by following an article on this URL” Rather than directing to the URL (which does not exist in the manuscript) it is better to provide a brief overview of how to do customization, and then refer the reader to the relevant article.

15. Abstract: The following sentences do not flow/read well “To measure task-level usability, one-way ANOVAs were conducted for each task separately to measure efficiency and effectiveness. However, for effectiveness, the touchscreen was significantly better in one task only. Therefore, an ANCOVA was conducted taking efficiency measure (Task completion time) as an independent variable and effectiveness (number of errors) as a covariate.” Please improve.

16. What is considered as an error in the calculation of effectiveness? Please define.

Other Comments:

1. Full forms of abbreviations be included if not already included elsewhere. For instance, SUS, ACM, and CHI.

2. L278, 279. “The problem was resolved”, “The dropout ratio was 10%” may be deleted as these are evident. Similarly, other redundant information may also be removed.

3. Repetitions should be avoided. For instance, “the screen was recorded” using Camtasia is repeated thrice in the paper.

4. The paper should be proofread well to improve readability and grammatical mistakes. Several sentences such as L24, L39-40, L47, L91-92, L102, L109-110, L222, L296, L359, L404, etc. should be improved for clarity. Common grammatical mistakes made in the paper involvement agreement of subject with verb.

6. PLOS authors have the option to publish the peer review history of their article (what does this mean?). If published, this will include your full peer review and any attached files.

Reviewer #1: No

Reviewer #2: No

---

## [Author Response · Author response to Decision Letter 0]

7 Nov 2023

Manuscript PONE-D-23-26336 Response to Reviewers 

Dear Editor, thank you for giving us the opportunity to submit a revised draft of the manuscript “Touch or Click Friendly: Towards Adaptive User Interfaces for Complex Applications” for publication in the PLOS One. We appreciate the time and effort that you and the reviewers dedicated to providing feedback on our manuscript and are grateful for the insightful comments on and valuable improvements to our paper. We have incorporated all the suggestions made by the reviewers. Those changes are highlighted within the manuscript with track changes. Please see below, point-by-point response to the reviewers’ comments and concerns. 

Reviewer 1

Comment 1:

Abstract

It is not clear how the recommendation in the last sentence follows from the objective of the study. The original question was about usability of the input device, but the recommendation is for a customized menu.

Response: 

We are thankful to the reviewer for pointing out the issue. There was already a sentence in the abstract in lines 11 and 12 with the text “The tasks were evaluated in terms of touch-friendly or click-friendly using efficiency, effectiveness, and satisfaction parameters to propose a customized user interface”. However, the term user interface is very generic and therefore, we have changed it to “customized task menu” to be precise. We hope this will align with the recommendation in the last sentence. 

Comment 2:

Introduction

First two paragraphs do not describe the purpose of the paper very well and lack direction. The introduction to the paper begins in earnest around line 43.

Earlier in the introduction there needs to be some explanation of “task” and “action” based analysis and methods, and what this study is intending to do. As written, those terms are not specific.

Response:

By agreeing with the reviewer, we have now removed the first two paragraphs and instead started the introduction by explaining the actions and the tasks. Please see section Introduction, lines 24 – 30. 

Comment 3:

The objective of this study is not made clear in the introduction. There is some potentially interesting exploration of existing research with the discussion around task classification, and the statements about challenges to this type of study are interesting. This needs further discussion in the introduction and a stronger, concise summary of how this study addresses the challenges. Some of this is detailed better in Section 2.

Response:

We are thankful to the reviewer for the suggestion. Reading again the introduction section in the context of the reviewer's comment, made us realize that the flow of the introduction does not go well. Therefore, we have deleted, rewritten, and reorganized various paragraphs and texts to make the introduction readable. We hope now the introduction is a lot clearer.

Comment 4:

Lines 106-107 – The objective of the study should be clearly stated, and not need to be inferred. This sentence could clearly summarize the objective/aim of the study.

Response:

We have now rewritten the text in the introduction. Furthermore, we have merged the sections in the related work into one as suggested by reviewer 2 and introduced a table as suggested by the reviewer 1. We hope these changes would have significantly improved the readability of the manuscripts’ first two sections.

Comment 5:

The term “performance” needs to be clearly defined, ideally in either the introduction or sections 2.1/2.2. The term “usability” also comes into section 2.2, and that is another that needs a quick definition. The measure is often time or error frequency, but it’s not clear what’s being considered “good performance” in these tasks. Especially as the tasks are identified as “cut, copy, paste…” it’s unclear what metric would be “better” for these tasks. This is important since this is a comparison of devices, and presumably, they all are capable of performing the task at some basic level.

Response:

We are thankful to the reviewer for identifying the shortcomings. We have now included relevant details at the start of the related work. We are hopeful that these details will make the section clearer and readable. Please see the related work section, lines 80-94.

Comment 6:

Section 2.3 – the demographic comparisons of previous studies could be better communicated as a table, and of interest will be the task types and what the measure of “performance” was. 

Response:

We have now merged sub-sections in the related work and introduced the table as Table 1. Please see the related work section (Table 1).

Comment 7:

Section 2.4 – starts discussing some of the metrics of previous studies, which is good, although “comfort metrics” can be expanded upon. Of interest is whether these were entirely user reported and whether that data was collected during or after the task. The statement in 169/170 is not supported by the preceding sentences, if it was a conclusion of the referenced study, that should be made more clear. The statement invites argument in this context because this study is stated to be examining an application that employs a combination of tasks, and it sounds awkward to have to switch between input devices for related operations in a word processor. 

Response:

We are thankful to the reviewer for the comment. As the reviewer 2 asked for a brief and meaningful related work, we have substantially reduced the related-work section. We have added a table as per reviewer 1 recommendation and briefly highlighted the related work. Therefore, the relevant section to the above comment is also removed. We are hopeful that reviewers will feel the new rewritten related work section is more objective, brief, and understandable.

Section 3 - Methods and Materials

Comment 8:

The objective and applicability of this study are still not clear. The description of the tasks selected doesn’t seem to align with any use case or practical application; the tasks were selected based on the number of input operations and their placement on a very information-heavy tool ribbon. The Tool ribbon in MS 365 groups tasks/operations (roughly) according to the phase of document production/review…not ease of access for novice users. This is also the first place in the study that “novice users” is brought up at all, and the relevance of that classification to the study outcomes is not apparent. When selecting the optimal tool for anything, there is typically a distinct set of criteria for novice and expert users. If the measurement of accuracy and efficiency in task completion is the “performance” the study is seeking, novice users do not seem to be the ideal subjects.

Response:

We are thankful to the reviewer for the comment. As the novice users were forward referenced, therefore created confusion for the reader. We have now moved the “Selection of the participant” sub-section to the beginning of the section to resolve forward referencing. This we hope would improve the readability. 

Furthermore, we added text (Page 7, lines 125- 130) to clarify the aim and objective of the research. 

A scenario document was provided to the participants for preparation as experimental task and as a practical application. For the related description please see sub-section 3.3 “Grouping of the tasks”. 

As already mentioned in the related work section, measurement metrics were efficiency and effectiveness along with user satisfaction. Therefore, we agree with the reviewer that novice users do not seem to be ideal subjects. Yet, for a generalize adaptable user interface we deem their inclusion fit. Furthermore, for a customized user menu, we did not conduct or compared three types of users with each other but taken data of all users together. Therefore, the selection of the novice users did not seem to affect the outcomes. 

Comment 9:

Section 3.5 – 268-270 sentence does not make sense. I think the term you are looking for might be “heterogeneous” as homogeneous data sounds like it would be more readily generalizable. This sentence might not be very useful in detailing the method of analysis.

Response: 

We are thankful to the reviewer for the correction. Incorrect wording is used as we meant to say the data collected from homogenous type of users. Therefore, the sentence is now corrected. Please see Page 7, lines 128 – 130. 

Comment 10:

It is also unclear whether the demographic divisions were determined a priori, and if so, why. Why was five years considered to be a threshold of relevance?

Response:

Yes, the demographics divisions were determined a priori. The reason was the availability of convenient sample. The details are provided on pages 7 & 8 lines 143-150. Similar subjective categorization is created by [5], [24], and [30]. Therefore, we created the categorization on the basis. This fact is explained on page 7, lines 130 – 133. 

Comment 11:

280-282 “no uncorrected eyesight or color blindness” doesn’t make sense. 

Response:

The text is corrected. Please see page 8, lines 150 – 152.

Section 4 – Experimental setup

Comment 12:

317-318 it is unclear how a break prevents or minimizes boredom. If no other activities or movement were allowed, it seems more likely that this break might have impacted fatigue? Intuitively it seems that a simple break might actually increase boredom, in the absence of a directed task.

Response:

The participants were free to do any activity during the break. The fact is now added in the place. Please see page 11, line 228.

Comment 13:

323-325 Who analyzed the videos and how was consistency/competence assured?

Response:

The required information is now added on page 11, lines 228 – 232.

Section 4 – Results 

Comment 14:

Please address the redundant section number

Response:

Redundant sections are now addressed.

Comment 15:

329 – if it is clear from research, please include a citation.

Response:

The text was a repetition. Therefore, we removed the first 3-4 lines to make the result section readable. 

Comment 16:

334-335 This is a lot of hypothetical and qualifying statements for a results section. It makes the communication unclear. 

Response: 

We are thankful to the reviewer for the comment. We observed that the text was a repetition of the aims of the research and was making the introduction to the results unreadable. Therefore, we have modified the introductory part and removed some text. The text to which the comment was referring was also removed. Please see first paragraph of the “Results” section.

Comment 17:

337-345 it is difficult to tell whether the ANOVA is appropriate in this case without a very clear statement of which variables are being analyzed and how those variables were being measured. The long quotation seems to suggest that a low level of significance was observed, but the authors do not state why that lower level is appropriate to this study.

Response:

The variables that are being analyzed and how the variables are being measured is explained in the last paragraph of section 4 (Page 11, lines 231 – 236). Furthermore, the long quotation is moved in sub-section 5.3 to the last paragraph. We think that this part would make more sense in the position. We have modified some text as well to make it more readable. Please see page 17,18 lines 305-312.

Comment 18:

Table 2 “Better” is a value judgment, a title that is more directly descriptive of what was measured in this study would be more clear in communicating results.

Response:

By agreeing with the reviewer, we have now replaced the word “better” with efficient and effective words.

Comment 19:

385-389 The description of variables here is good, but the opening statement, that this leads to “better customization” is very confusing and not linked to the stated aims.

Response:

By agreeing with the reviewer, we have now changed “better customization” to “A One-Way ANCOVA was conducted to suggest such customization of menus that considers both efficiency and effectiveness”. Please see page 15, lines 282, 283.

Comment 20:

404-405 is troubling because it looks like the analysis method was selected. 

Response:

The wording of the sentence was erroneous. The erroneous sentence “As can be seen from Table 5, An ANOVA for efficiency and an ANCOVA for efficiency with efficiency as a covariate resulted in some changes” is now replaced with the sentence “Table 6 shows changes between ANOVA conducted for measure efficiency and an ANCOVA conducted for measure efficiency with effectiveness data as a covariate”. Please see page 18, lines 295, 296.

Comment 21:

421-429 the specific variables and relationships need to be summarized more concisely, and without extensive qualifying statements somewhere. It remains unclear why this statistical analysis is relevant to any conclusion for the study.

Response:

We have modified the text and introduced a summary table for the user satisfaction statistics. We hope the text will be readable and comprehensible. Please see pages 18, Table 7 along with corresponding text. 

Comment 22:

There are interesting elements in this study and the data set and analysis can add to this field, but major revision to the presentation would help to frame the contribution and make the practical application for these findings more evident.

Response:

We are very thankful to the reviewer for the critical analysis that has resulted in an improved version of the manuscript. 

Reviewer 2

The paper evaluates the usability of different input devices such as mouse, touchpad, and touchscreen in performing a wide range of tasks in MS Word. The task-level usability is evaluated in terms of parameters that include efficiency, effectiveness and user satisfaction in line with ISO-9241-11 standard recommendations. Accordingly, the tasks have been categorized into touch-friendly and click-friendly. Further, the paper considers participant groups with multiple levels of expertise to ensure generalizability of results. The tasks considered are also of mixed difficulty levels. The work has the potential to pave the way for improving user interface design of complex applications by grouping tasks based on the most suitable input device. However, the paper can only be considered for publication after the following comments are thoroughly addressed.

Response: We are thankful to the reviewer for investing time to improve this manuscript. We are also thankful to the reviewer for realizing the potential in this work and to suggest changes that resulted in the improvement of the manuscript. Below we have addressed all the comments of the reviewer. 

Comment 1:

The authors should add more recent literature in Sections 1 (Introduction) and 2 (Related Work).

Response:

We have added following 4 new references from 2021 to 2023 and removed 7 old references. 

 [1] Rogers, H., Chalil Madathil, K., Joseph, A., McNeese, N., Holmstedt, C., Holden, R., & McElligott, J. T. (2021). Task, usability, and error analyses of ambulance-based telemedicine for stroke care. IISE Transactions on Healthcare Systems Engineering, 11(3), 192-208.

[2] Wu, J., Zhu, Y., Fang, X., & Banerjee, P. (2023). Touch or click? The effect of direct and indirect human-computer interaction on consumer responses. Journal of Marketing Theory and Practice, 1-16.

[4] Chakraborty, P., & Yadav, S. (2023). Applicability of Fitts’ law to interaction with touchscreen: review of experimental results. Theoretical Issues in Ergonomics Science, 24(5), 532-546

[14] Y. Rogers, H. Sharp, and J. Preece, (2023). Eds., Interaction Design: Beyond Human-Computer Interaction, 6th Ed., New York, NY, USA: John Wiley & Sons Ltd, 2023.

 Comment 2:

The way the existing literature has been reported in Related Work needs to be improved. It would be helpful if conclusions/insights derived from those studies are reported (briefly) so that the reader benefits from reading it. Just reporting the number/gender of participants, number of tasks, and what has not been done in the previous studies is not much helpful.

Response: We have now significantly revised the related work section and reduced it to about 1100 words from over 2000 words by summarizing it. In addition, we also added a table as per the reviewer 1 suggestion. 

Comment 3:

The authors may want to include inferences drawn from studies [9], [17], [24] as these studies have considered mouse, touchscreen, and touchpad comparison [as depicted from Table 1]. Further, how these conclusions relate/align/ with the findings of this paper?

Response:

We have now discussed the inferences from the studies [9], [17], [24] which are now [2], [18], and [27]. Please see page 5, line 112- 118.

Comment 4:

Line 238: …, all tasks were assessed as per the GOMS method. Assessed for what? L239-L244 may be suitably modified to make it evident and highlight its usefulness (conceptually).

Response:

The GOMS model is used to describe a user’s cognitive structure on four components. The Goals (G), Operators (O), Methods (M) and Selections (S) used in the tasks as per GOMS model is described. We have now modified the text to highlight its usefulness. Please see Page 9, lines 172-180.

Comment 5:

Was there any specific reason that MS Word 2010 was used?

Response:

MS Word 365 was used. The reason was that it was the most recent version of the MS Word. The erroneous wording is now removed from sub-section 3.6 i.e., “Equipment”. 

Comment 6:

A schematic of the whole experimental protocol would be more beneficial than Figure 1 (which currently includes a picture of a laptop).

Response:

Old Figure 1 is now replaced with a schematic of the experimental protocol.

Comment 7:

L339-L341: “A compromise power analysis for (N=60, df =2, No. of Groups=3, effect size= Medium) revealed a power of 0.73.” What does this signify/imply?

Response:

The analysis was not suitable for the study and the text was not clear. Therefore, the related text is removed from the manuscript to improve the readability of the paper.

Comment 8:

There is a mismatch between the title of Section 4.3 (L384) and L387. While the title mentions efficiency as a covariate, the text in L387 considers effectiveness as a covariate. Also, the title of Section 4.3 is a bit confusing and may be modified suitably. Further, what is the rationale for the choice of a particular covariate measure?

Response: It was a typing mistake. We have now corrected the mistake. The title and the consecutive relevant text were “Efficiency with effectiveness as a covariate”.

Comment 9:

Table 5 reports numbers. It may be more useful if the numbers are reported as percentages so as to make interpretation more efficient. Also, in the main text of the “Results” section, the percentages should also be provided consistently for ease of interpretation.

Response: Table 5, now changed to Table 6 is modified as per the reviewer's instructions. Furthermore, the relevant text is also modified accordingly for ease of interpretation.

Comment 10:

If possible, the authors may also want to report the summarized results of the efficiency and effectiveness of various input devices for different expertise groups. This will help understand whether there is any impact of expertise on the usability of devices, at least qualitatively.

Response:

We have added the summarized results of the efficiency and effectiveness for different expertise groups in Appendix D (attached with the main manuscript).

Comment 11:

Section 4.5. Please improve for better clarity, especially the sentences related to validity. Also provide a conceptual significance of Inter-rater reliability and internal consistency, or at least refer the reader to a suitable reference.

Response:

The text is now revised. Furthermore, a reference highlighting the importance of reliability is also included. Please see sub-section 5.5 “Validity and Reliability Analysis”.

Comment 12:

L434-435: “Furthermore, a covariate analysis is conducted as explained in subsection J of Section IV to evaluate the potential confounding factors affecting the tasks” Subsection J does not exist.

Response:

The error is now corrected to “Furthermore, a covariate analysis (c.f. subsection 5.3) is conducted to evaluate the potential confounding factors affecting the tasks.” 

Comment 13:

L476: “with a touch-to--to-mouse/touchpad number 3 vs. 9 or with a ratio of 1:3” is not clear.

Response:

The text is modified. Please see page 21, lines 370-373.

Comment 14:

L 478: “The customization can be done as per the suggestion by following an article on this URL” Rather than directing to the URL (which does not exist in the manuscript) it is better to provide a brief overview of how to do customization, and then refer the reader to the relevant article.

Response:

The missing URL is now added. In addition, a summary about customization is added. Please see page 21, lines 373-376.

Comment 15:

Abstract: The following sentences do not flow/read well “To measure task-level usability, one-way ANOVAs were conducted for each task separately to measure efficiency and effectiveness. However, for effectiveness, the touchscreen was significantly better in one task only. Therefore, an ANCOVA was conducted taking efficiency measure (Task completion time) as an independent variable and effectiveness (number of errors) as a covariate.” Please improve.

Response:

We have now updated the text with the following text “To assess task-level usability, individual one-way ANOVAs were performed for each task to gauge both efficiency and effectiveness. It's worth noting that the touchscreen significantly outperformed other input methods in just one specific task regarding effectiveness. Consequently, an ANCOVA was employed, with task completion time as the independent variable and the number of errors as a covariate, to further investigate effectiveness.”

Comment 16:

What is considered an error in the calculation of effectiveness? Please define.

Response:

The calculation of error is described on page 11, lines 233-234. 

Other Comments:

Comment 17:

Full forms of abbreviations be included if not already included elsewhere. For instance, SUS, ACM, and CHI.

Response:

The corresponding abbreviations are now added. 

Comment 18:

L278, 279. “The problem was resolved”, “The dropout ratio was 10%” may be deleted as these are evident. Similarly, other redundant information may also be removed.

Response:

The specific text mentioned in the comment is removed. Moreover, we have removed some other redundant information as well.

Comment 19:

Repetitions should be avoided. For instance, “the screen was recorded” using Camtasia is repeated thrice in the paper.

Response:

The repetition is now removed, and the text accordingly modified.

Comment 20:

The paper should be proofread well to improve readability and grammatical mistakes. Several sentences such as L24, L39-40, L47, L91-92, L102, L109-110, L222, L296, L359, L404, etc. should be improved for clarity. Common grammatical mistakes made in the paper involvement agreement of subject with verb.

Response:

All the corresponding text is corrected or was already deleted in response to the earlier comments of both reviewers. Furthermore, we have carefully proofread the manuscript for further grammatical mistakes and corrected them. We hope that the manuscript will be readable as compared to the earlier version.

---

## [Decision Letter · Decision Letter 1]

29 Nov 2023

PONE-D-23-26336R1Touch or Click Friendly: Towards Adaptive User Interfaces for Complex ApplicationsPLOS ONE

Dear Dr. Khan,

Thank you for submitting your manuscript to PLOS ONE. After careful consideration, we feel that it has merit but does not fully meet PLOS ONE’s publication criteria as it currently stands. Therefore, we invite you to submit a revised version of the manuscript that addresses the points raised during the review process.

We look forward to receiving your revised manuscript.

Kind regards,

Rajagopalan Srinivasan

Academic Editor

PLOS ONE

Reviewers' comments:

Reviewer's Responses to Questions

**Comments to the Author**

1. If the authors have adequately addressed your comments raised in a previous round of review and you feel that this manuscript is now acceptable for publication, you may indicate that here to bypass the “Comments to the Author” section, enter your conflict of interest statement in the “Confidential to Editor” section, and submit your "Accept" recommendation.

Reviewer #1: (No Response)

Reviewer #2: (No Response)

2. Is the manuscript technically sound, and do the data support the conclusions?

Reviewer #1: Yes

Reviewer #2: Yes

3. Has the statistical analysis been performed appropriately and rigorously? 

Reviewer #1: Yes

Reviewer #2: Yes

4. Have the authors made all data underlying the findings in their manuscript fully available?

Reviewer #1: Yes

Reviewer #2: Yes

5. Is the manuscript presented in an intelligible fashion and written in standard English?

Reviewer #1: Yes

Reviewer #2: Yes

6. Review Comments to the Author

Reviewer #1: I have included detailed comments in an attached file. Minor revisions are suggested, to clarify new figures and add a clarification statement.

Reviewer #2: The authors have significantly improved the manuscript. However, the following comments need to be addressed before it can be accepted for possible publication.

Related Work

The insights/conclusions that can be derived from the existing literature are still missing and need to be included. Currently, these are missing. For instance, information like which input device is better for what type of tasks? It would be beneficial to the reader if a summary of the same is also included in Table 1.

The authors seem to have misinterpreted the reviewer's previous comment on this which read: “The way the existing literature has been reported in Related Work needs to be improved. It would be helpful if conclusions/insights derived from those studies are reported (briefly) so that the reader benefits from reading it. Just reporting the number/gender of participants, number of tasks, and what has not been done in the previous studies is not very helpful.” The reviewer meant that the findings/conclusions from the previous studies should at least be included briefly.

Experimental Setup

Fig. 1 needs to be improved significantly so that the experimental methodology is clear from the start to the data analysis. There is a lot of unnecessary information in Fig. 1, such as scenario text copied to desktop, participants invited to college. Make it crisp. The current figure is nothing more than a summary of text that is already there in Section 4.

Section 5.1

“As Table 3 shows, out of fifty-six cases, 27 or 48% of the tasks’ execution time was less with a mouse, while 21 (37%) tasks were executed with significantly less execution time. Furthermore, 3 tasks had a large effect size (η2 ≥ 0.14), 8 tasks had a medium effect size (η2 ≥ 0.06), and 10 tasks had a small Effect size (η2 ≥ 0.02).” Please write explicitly whether these 21 tasks are out of the 27 tasks mentioned earlier. Also, what do the effect sizes signify here need to be stated.

Section 5.3

What is the rationale for choosing Efficiency with effectiveness as a covariate and not the other way around i.e. Effectiveness with efficiency as a covariate?

While Table 6 now reports the results in percentages, the text still contains numbers. Reporting of percentages along with numbers in the main text will make it more readable.

7. PLOS authors have the option to publish the peer review history of their article (what does this mean?). If published, this will include your full peer review and any attached files.

Reviewer #1: No

Reviewer #2: No

---

## [Author Response · Author response to Decision Letter 1]

3 Dec 2023

The response to the reviewers is attached in the file named as "Response to reviewers PONE-D-23-26336_R2".

---

## [Decision Letter · Decision Letter 2]

28 Dec 2023

Touch or Click Friendly: Towards Adaptive User Interfaces for Complex Applications

PONE-D-23-26336R2

Dear Dr. Khan,

We’re pleased to inform you that your manuscript has been judged scientifically suitable for publication and will be formally accepted for publication once it meets all outstanding technical requirements.

Kind regards,

Rajagopalan Srinivasan

Academic Editor

PLOS ONE

Additional Editor Comments (optional):

Reviewers' comments:

Reviewer's Responses to Questions

**Comments to the Author**

1. If the authors have adequately addressed your comments raised in a previous round of review and you feel that this manuscript is now acceptable for publication, you may indicate that here to bypass the “Comments to the Author” section, enter your conflict of interest statement in the “Confidential to Editor” section, and submit your "Accept" recommendation.

Reviewer #2: All comments have been addressed

2. Is the manuscript technically sound, and do the data support the conclusions?

Reviewer #2: Yes

3. Has the statistical analysis been performed appropriately and rigorously? 

Reviewer #2: Yes

4. Have the authors made all data underlying the findings in their manuscript fully available?

Reviewer #2: Yes

5. Is the manuscript presented in an intelligible fashion and written in standard English?

Reviewer #2: Yes

6. Review Comments to the Author

Reviewer #2: The authors have addressed the comments. They have significantly improved the manuscript. I have no further comments.

7. PLOS authors have the option to publish the peer review history of their article (what does this mean?). If published, this will include your full peer review and any attached files.

Reviewer #2: No

---

## [Editor Report · Acceptance letter]

24 Jan 2024

PONE-D-23-26336R2 

PLOS ONE

Dear Dr. Khan, 

I'm pleased to inform you that your manuscript has been deemed suitable for publication in PLOS ONE. Congratulations! Your manuscript is now being handed over to our production team.

Kind regards, 

on behalf of

Dr. Rajagopalan Srinivasan 

Academic Editor

PLOS ONE